# Teachers’ Emotional Intelligence and Organizational Commitment: A Moderated Mediation Model of Teachers’ Psychological Well-Being and Principal Transformational Leadership

**DOI:** 10.3390/bs14040345

**Published:** 2024-04-20

**Authors:** Mingwei Li, Feifei Liu, Chuanli Yang

**Affiliations:** 1Business School, Beijing Normal University, Beijing 100875, China; mingweili@bnu.edu.cn; 2School of Teacher Education, Nanjing Normal University, Nanjing 210023, China; 3School of Education Science, Nanning Normal University, Nanning 530001, China; yangchuanli@nnnu.edu.cn

**Keywords:** teachers’ organizational commitment, emotional intelligence, psychological well-being, principal transformational leadership

## Abstract

Given the global challenge of increasing teacher attrition and turnover rates, the exploration of factors and mechanisms that improve teachers’ organizational commitment has become a pivotal topic in educational research. In this context, the present study examines the influence of teachers’ emotional intelligence on their organizational commitment, with a specific inquiry into the mediating role of teachers’ psychological well-being and the moderating role of principal transformational leadership, as informed by the broaden-and-build theory of positive emotions and the trait activation theory. We verified this study’s hypotheses based on 768 valid questionnaires collected from Chinese primary and secondary school teachers. The results reveal that teachers’ emotional intelligence can predict their organizational commitment both directly and indirectly through the mediating role of psychological well-being. Additionally, principal transformational leadership amplifies the positive effect of teachers’ emotional intelligence on psychological well-being and, subsequently, organizational commitment. These findings theoretically deepen our understanding of the psychological pathways and the boundary conditions linking teachers’ emotional intelligence to their organizational commitment, while also offering valuable practical implications for building a stable and effective teaching workforce.

## 1. Introduction

Contemporary teachers confront a complex and dynamic reality characterized by various demanding situations, such as high-stakes testing, excessive workloads, limited professional autonomy, and scrutiny from parents and society. These challenges generate excessive work pressure, erode job satisfaction and well-being, and lead to a global, severe issue of increasing turnover and attrition rates [1]. For example, a national survey in China revealed that nearly 62 percent of over 10 million full-time primary and secondary school teachers have contemplated quitting their jobs [2]. Additionally, according to the Teaching and Learning International Survey (TALIS 2018), it is estimated that almost half of new teachers abandon the profession within five years [3]. Consequently, enhancing the motivational basis of teachers’ attitudes toward work and strengthening their organizational commitment (OC) has emerged as a central topic in educational research worldwide [4,5].

OC is defined as an individual’s identification with, attachment to, and involvement in a particular organization, encompassing three dimensions: affective, normative, and continuance [6,7]. In the educational context, OC reflects teachers’ sense of fidelity and psychological attachment to their schools [8,9]. Committed teachers align with the school’s cultures and goals, feel a sense of obligation, exert extra effort in their work, and possess a strong desire to remain with their current school [10]. Previous studies have linked teachers’ OC with positive outcomes at both the individual and organizational levels, including enhanced teacher work engagement [11], organizational citizenship behavior [12], job performance, school effectiveness [13], and reduced turnover intentions [10]. Thus, exploring the factors and mechanisms influencing teachers’ OC is imperative for developing a stable and high-quality teaching workforce, enabling educational institutions to fulfill the requirements of compulsory education and benefit all students.

Recent studies have highlighted several individual factors contributing to teachers’ OC, among which emotional intelligence (EI) has gained increasing attention [4,14,15,16]. EI was first introduced by Salovey and Mayer in 1990 as “the ability to monitor one’s own and others’ feelings and emotions, to discriminate among them and to use this information to guide one’s thinking and actions” [17]. Since then, EI has emerged as a focal topic in research and practice, especially following the “affective revolution” of the last few decades [18], which has emphasized EI’s critical role in work-related outcomes across various organizational contexts, including education [19]. Specifically, emotions are central to both teaching and teachers’ lives, making the profession an emotionally driven one frequently requiring emotional labor [20]. Therefore, EI, the capacity to attend to, process, and act upon emotional information [17,21,22,23], is positively linked to teachers’ attitudinal and behavioral outcomes, such as increased job satisfaction [19], improved job performance [24], and reduced burnout and stress [25].

Focusing on OC, prior research indicates that individual differences in EI significantly predict employees’ commitment to their organizations [26,27]. Highly emotionally intelligent individuals, capable of perceiving and regulating emotions and using emotional information to guide thinking and behavior, are more prone to establishing strong emotional bonds with their organization, aligning with its norms and goals, and recognizing the costs associated with leaving, thereby exhibiting stronger OC than their less emotionally intelligent counterparts. While empirical studies have demonstrated the positive relationship between EI and OC, including those specifically addressing teachers in the school context [15,28], the psychological pathways and boundary conditions of this consequential relationship remain largely undetermined. This study aims to bridge this gap by investigating the mediating role of psychological well-being (PWB) and the moderating role of principal transformational leadership (PTL), offering insights for educational practitioners to enhance teachers’ OC through improved teacher development initiatives and leadership training programs.

### 1.1. Teachers’ Psychological Well-Being as a Mediator

We contend that EI can positively predict OC by enhancing psychological well-being (PWB)—a state characterized by a sense of happiness [29]. Rooted in the eudemonic tradition, PWB signifies optimal psychological functioning and the realization of personal potential [30]. Theory positions EI as an important antecedent of PWB. Intrapersonally, self-emotion appraisal and the effective utilization and regulation of emotions aid individuals in managing stress and mitigating negative feelings. Interpersonally, EI-related emotional awareness and regulation processes enhance one’s ability to interact with others, resolve social conflicts, and foster social networks. Consequently, individuals with high levels of EI are more likely to sustain positive mental states, achieve personal aspirations, and find value and meaning in life [21,31,32,33], culminating in enhanced PWB. This relationship is particularly salient in the teaching profession, where teachers frequently encounter stress and engage in emotion-demanding interactions with students, parents, and colleagues. Empirical studies consistently confirm a positive correlation between EI and healthy indicators of well-being [32,34] and a negative correlation with anxiety, depression, perceived stress, and burnout [35] among teachers.

PWB can have a positive implication on OC. According to the broaden-and-build theory of positive emotions [36,37], positive emotions (e.g., the experience of PWB) can expand one’s repertoire of thoughts and actions, which, in turn, can foster enduring personal resources, leading to positive attitudes, cognitions, and behaviors. When applied to the workplace, the manifestation of these positive emotions can transform employees’ perceptions of their work, viewing it as a “calling” rather than merely a job [38,39]. Consequently, employees build emotional bonds with their organizations, strongly identify with organization goals, and express a desire to remain part of the organization [40,41,42], all of which constitute key elements of OC. While empirical studies outside the realm of education have confirmed a positive association between employees’ PWB and OC [43,44], research focusing specifically on the relationship within the school context remains limited, albeit with notable exceptions. For example, by collecting data from 346 high school teachers, Heidari et al. found that differences in teachers’ PWB could explain the variations in their OC [45]. Synthesizing these arguments, it stands to reason that more emotionally intelligent teachers—who adeptly perceive, process, and manage their own emotions and those of others—will experience higher levels of PWB, and the positive emotions prompted by PWB, in turn, will enhance their commitment to their respective schools. Therefore, we posit the following:

**Hypothesis** **1.**
*Teachers’ emotional intelligence predicts organizational commitment through the mediating role of psychological well-being.*


### 1.2. Principal Transformational Leadership as a Moderator

Do individual differences in EI directly affect and determine one’s cognitions, attitudes, and behaviors? Not necessarily. Emerging research on the effect of individuals’ EI demonstrates the joint influence of environmental variables, which act as critical boundary conditions or, simply put, moderators [46,47]. A prominent environmental attribute in the organizational context is transformational leadership, which signifies a typical kind of effective leadership characterized by four behavioral components: inspirational motivation, idealized influence, intellectual stimulation, and individualized consideration [9,48,49]. In school organizations, transformational leadership entails principals providing care and support for teachers, stimulating their inner motivations and potential, and assisting collaborative efforts toward school goals [50,51]. This leadership style, characterized by its ethical, visionary, and relational attributes, is linked to multiple desired outcomes within and beyond the school organization context, including positive organizational climate, employees’ trust in leaders, and high levels of employees’ psychological capital, well-being, satisfaction, and commitment [9,52,53,54].

Given the prominent positive effect of transformational leadership, it may interact with EI to jointly influence employees’ PWB and OC. The trait activation theory [55] provides a theoretical framework to elucidate this interacting effect. It posits that EI, conceptualized as a constellation of emotional perceptions within the trait model and assessed via self-report questionnaires [22], is akin to a lower-order personality trait requiring trait-relevant cues from the environment for expression. Specifically, a trait is more likely to manifest in amenable situations that signal the appropriateness and necessity of such expression [55]. In schools where teachers are supported by transformational principals, who set high expectations and instill confidence in teachers’ interests and abilities, they are more likely to feel secure and inspired to behave in an emotionally intelligent way. There is a better chance for them to skillfully manage stress and negative emotions, maintain a healthy psychological state, and foster a strong commitment to the school. In essence, principal transformational leadership (PTL) may activate teachers’ EI and strengthen its positive influence on PWB and OC. Conversely, in the absence of principals’ transformational behavior, teachers may feel unsupported and less motivated to show their capabilities of perceiving, interpreting, and regulating emotions in themselves and others, diminishing its positive impact. That is, when teachers are in schools with low PTL, even emotionally intelligent teachers may struggle to achieve the same levels of PWB and OC as their counterparts in schools with high PLT. Therefore, we hypothesize as follows:

**Hypothesis** **2.**
*Principal transformational leadership moderates the relationship between teachers’ emotional intelligence and psychological well-being.*


**Hypothesis** **3.**
*Principal transformational leadership moderates the relationship between teachers’ emotional intelligence and organizational commitment.*


Moreover, according to previous arguments [56,57], when the first half or second half of the mediation pathway is moderated by the moderator variable, the mediation effect itself can also be moderated. Therefore, we propose the following:

**Hypothesis** **4.**
*Principal transformational leadership moderates the mediating effect of psychological well-being in the relationship between teachers’ emotional intelligence and organizational commitment.*


### 1.3. The Hypothesized Model of Study

To sum up, this study aims to examine the psychological pathway and the boundary conditions of the relationship between teachers’ EI and their OC. The hypothesized model of the study is depicted in Figure 1.

## 2. Methods

### 2.1. Participants and Procedure

Convenience sampling was used in this study. With the assistance of local education authorities in two cities in mid-east China’s Jiangsu and Anhui provinces, 12 public primary and secondary schools were randomly invited for participation. Prior to the formal questionnaire survey, invitation letters were sent to school administrators by researchers to obtain consent from the teachers. The letter explicitly outlined the survey’s nature, purpose, and methods, emphasizing voluntary participation and ensuring the confidentiality and anonymity of the responses. Teachers who agreed to participate completed the questionnaire independently in their offices during visits from our specialized personnel and returned their questionnaires on-site. It took about 10 min to answer all the questions. All participants were informed of their right to withdraw at any time and encouraged to pose any questions regarding the survey.

In total, 880 questionnaires were distributed and 834 were retrieved. After data cleaning, 768 valid questionnaires were obtained (effective response rate = 87.3%) and then used for data analysis. Among all the participants, there were 127 males (16.5%) and 641 females (83.5%). Regarding the professional ranking, 20.8% of the participants held a senior professional rank, whereas 79.2% possessed a primary or secondary one. As for teaching experience, the distribution was as follows: 19.8% had 1–5 years, 15.4% had 6–10 years, 28.5% had 11–20 years, and 36.3% had over 20 years of experience. The educational background of the participants varied, with 14.5% holding a college or high-school degree, 83.1% having a Bachelor’s degree, and 2.5% possessing a Master’s degree.

### 2.2. Measures

#### 2.2.1. Teachers’ Emotional Intelligence

Teachers’ emotional intelligence was measured using the sixteen-item Wong Law Emotional Intelligence Scale (WLEIS) [58]. The scale comprises four dimensions: self-emotional appraisal (SEA), others’ emotional appraisal (OEA), regulation of emotion (ROE), and use of emotion (UOE). Responses were rated on a 5-point Likert scale, with higher average scores indicating a higher level of emotional intelligence. Sample items include “I am able to control my temper and handle difficulties rationally”, and “I have a good understanding of my own emotions”. The scale has shown adequate reliability and validity with Chinese teachers [19]. The Cronbach’s alpha coefficient of the scale was 0.926 in this study.

#### 2.2.2. Teachers’ Psychological Well-Being

The subscale of the psychological well-being of Zheng et al.’s employee well-being scale [59] was used in the current study to measure teachers’ psychological well-being. This sub-scale comprises 7 items, each rated on a 7-point Likert scale, examples of which include “I feel I have grown as a person”, and “I generally feel good about myself, and I’m confident”. Responses to the seven items were aggregated and averaged, with higher scores indicating a higher level of psychological well-being. The scale has been widely used in Chinese populations and has demonstrated good validity and reliability for Chinese teachers [60]. The Cronbach’s alpha coefficient of the seven items was 0.948 in the current study.

#### 2.2.3. Teachers’ Organizational Commitment

Teachers’ organizational commitment was measured using the Chinese version [61] of Meyer et al.’s original scale of organizational commitment [62]. The scale comprises nineteen items measuring affective, normative, and continuance commitment dimensions on a 5-point Likert scale. Sample items include “I am proud to tell others that I am a part of this school” (affective commitment), “I hope everyone should be loyal to their school and fulfill all their obligations” (normative commitment), and “Once I leave my current school, I will lose a lot of welfare benefits” (continuance commitment). Higher scores indicate a stronger commitment to the school. The scale demonstrated good reliability (Cronbach’s alpha coefficient = 0.873).

#### 2.2.4. Principal Transformational Leadership

We used the 26-item Chinese version of the transformational leadership scale in the current study [63]. The scale was adapted specifically for the context of school organizations to measure principals’ transformational leadership behavior. The scale contains four dimensions: (1) inspirational motivation, consisting of six items; (2) idealized influence, consisting of eight items; (3) intellectual stimulation, consisting of six items; and (4) individualized consideration, consisting of six items. Sample items include “Our principal clearly communicates the goals and priorities of the school”, “Our principal cares about the work, life, and growth of teachers, and sincerely provides suggestions for their development”, and “Our principal focuses on creating conditions for teachers to make full use of their strengths”. Teachers responded to the items on a 5-point Likert scale based on their perceptions of the principal’s behavior, with higher scores indicating higher principal transformational leadership. The Cronbach’s alpha coefficient of the scale was 0.956, showing good reliability.

### 2.3. Data Analysis

The current study used SPSS 26.0 and Mplus 8.3 for data analysis. First, descriptive statistics and correlation analysis were conducted by SPSS. Second, confirmatory factor analysis (CFA) was conducted by Mplus to examine the measurement model. Then, Hayes’ [64] macro program PROCESS in SPSS was used to test the mediation, moderation, and moderated mediation effect. During the analysis, the bias-corrected percentile Bootstrap method with 95% bias-corrected confidence intervals (CI) was used. A total of 5000 samples were randomly selected from the original dataset repeatedly with effects deemed significant if the CI did not contain zero.

As all of the data were collected through teachers’ self-reported questionnaires in this study, there could be Common Method Biases (CMBs). To address this concern, procedural control was initially applied as extensively as possible. Specific practices include using mature measurement scales, incorporating reverse-coded items, and randomizing the order of scale questions to prevent alignment with the central variable sequence, etc. Furthermore, we conducted Harman’s single factor test of common method bias. The result showed that the overall variation explained (41.33%) was less than 50% when all components were loaded into one general factor, suggesting that the common method bias did not significantly impact the study’s results [65].

## 3. Results

### 3.1. Test of the Measurement Model

We conduct CFA on the four-factor measurement model (i.e., EI, PWB, PTL, and OC). The results revealed a good fit to the data: χ^2^ = 448.77, d*f* = 129, RMSEA = 0.06, CFI = 0.95, TLI = 0.95, and SRMR = 0.05. We then compared this baseline model with three theoretically plausible alternative models (see Table 1). Since the model fits worsened as more variables were combined, discriminant validity was confirmed.

### 3.2. Descriptive Statistics and Correlation

Descriptive statistics and Pearson correlation coefficients for the study variables are presented in Table 2. EI was positively correlated with PWB, PTL, and OC; PWB was positively correlated with PTL and OC; and PTL was positively correlated with OC.

### 3.3. Hypothesis Testing

First, the mediating role of PWB was tested using Model 4 in PROCESS. As illustrated in Table 3, EI had a significant effect on PWB (β = 0. 81, *p* < 0.001) and PWB significantly affected OC (β = 0. 19, *p* < 0.001). The predictive effect of EI on OC remained significant when PWB was included (β = 0. 28, *p* < 0.001). The bootstrap 95% confidence interval for the direct and mediating effect was [0.20, 0.36], [0.10, 0.21], respectively, none of which included 0 (see Table 4), indicating that EI not only directly predicted OC, but also predicted OC through the mediating effect of PWB. Table 4 provides further details on the total effect and the proportion of direct and mediating effects.

Second, the mediated model with PTL as the moderator was tested using Model 8 in PROCESS. The results (see Table 5) showed that the interaction of EI and PTL significantly affected PWB (β = 0. 27, *p* < 0.001). This suggested that PTL moderated the relationship between EI and PWB, confirming Hypothesis 2. As displayed in Figure 2, simple slope analysis revealed that the positive effect of EI on PWB was stronger under high PTL (M + 1 SD, simple slope = 0.75, *p* < 0.001) compared to low PTL (M − 1 SD, simple slope = 0.44, *p* < 0.001). However, the predictive effect of the interaction of EI and PTL on OC was not significant (β = 0. 01, *p* > 0.05), leading to the rejection of hypothesis 3. Further, as shown in Table 6, PROCESS analysis showed that the indirect effect of EI on OC via PWB was stronger when PTL was high (M + 1 SD) rather than low (M − 1 SD). The index of moderated mediation was significant (estimate = 0.04, boot se = 0.02, 95% CI [0.0014, 0.07]). Thus, hypothesis 4 was supported.

## 4. Discussion

Highly qualified and committed teachers are the most important asset of schools [66]. However, teacher attrition has become a significant challenge confronting schools worldwide [1,67]. Therefore, exploring the factors and mechanisms to enhance teachers’ OC and retain high-quality teachers has emerged as a central topic in educational research. This study focuses on the influence of teachers’ EI on OC, alongside systematically examining the effect of PWB and PTL. Based on previous studies and drawing from the broaden-and-build theory of positive emotions and the trait activation theory, this study constructs a moderated mediation model with PWB as the mediating variable and PTL as the moderating variable. Our model elucidates how teachers with higher EI exhibit a stronger commitment to the schools they serve (the mediating role of PWB) and also addresses the question of under what school conditions the positive influence of teachers’ EI on PWB and OC is more significant (the moderating role of PTL). The research results contribute to the advancement of our understanding of psychological pathways and boundary conditions in the relationship between teachers’ EI and their OC, thereby offering valuable insights for educational practitioners aiming to build a stable and effective teaching workforce.

First, consistent with prior research [15,26], our findings revealed that teachers’ EI directly predicted their commitment to the schools they serve. Given the nurturing nature of the teaching profession, teachers engage in intensive face-to-face interactions with students every day, necessitating a considerable amount of emotional labor. Moreover, challenges such as the phenomenon of “accountability”, daily overexertion, no time, and classroom management exacerbate emotional demands on teachers, creating substantial pressure and potentially leading to resignations after emotional wear and tear [68]. The results of this study showed that teachers with high EI, who are proficient in recognizing, identifying, and understanding both their own and others’ emotions, as well as in appropriately expressing and managing negative emotions, and in establishing and sustaining positive interpersonal relationships, were shown to be more committed to their schools, compared to those who had difficulties in these aspects of EI. This validates EI as a pivotal personal resource in teaching work, reinforcing its growing emphasis in scholarly discussions [19,25].

Second, our study also indicated that PWB partially mediated the predictive effect of teachers’ EI on OC, extending the body of evidence demonstrating direct associations between EI and PWB [31,32,69], as well as PWB and OC [45]. This finding shed light on the mediating role of PWB between teachers’ EI and OC, aligning with the argument of the broaden-and-build theory of positive emotions [36]. Specifically, it implies that teachers with higher EI are more committed to their schools, partially due to their enhanced psychological functioning and more consistent positive emotional experiences. In essence, teachers’ superior EI-related skills facilitate higher levels of PWB, and these positive emotions prompted by PWB, in turn, raise their OC. This finding is also in line with the argument by Yin et al., who found, through longitudinal research, that teachers’ PWB significantly amplifies their commitment levels, although no reciprocal relationship between the two variables was identified [70]. Our recognition of the indirect pathway through PWB helps bridge the research gap on the intervening mechanisms between teachers’ EI and OC, thereby deepening our understanding of how emotionally intelligent teachers can stay highly committed to their schools.

Third, this study demonstrated that PTL positively moderated the relationship between teachers’ EI and PWB. Specifically, resonant with the trait activation theory [55], higher levels of reported PLT among teachers correspond to an increased activation of emotion-related capabilities through relevant cues, thereby facilitating the maintenance of high levels of PWB. That is to say, the positive effect of teachers’ EI on PWB is enhanced in schools with transformational leaders. In contrast, lower levels of reported PLT among teachers are associated with feelings of diminished care and support, resulting in less frequent experiences of positive emotions, and thus attenuating the positive role of EI in reaching high levels of PWB. The result echoes previous studies and arguments on the importance of environmental conditions in linking employees’ EI to individual outcomes [24,46,47], as well as the prominent role of transformational leadership in school settings [71], establishing PTL as an important boundary condition for the effect of teachers’ EI on PWB. Transformational school leaders, being highly considerate, visionary, and encouraging, increase the likelihood of teachers experiencing positive emotions, feeling personal accomplishment, and maintaining psychological health [9,49]. According to Arnold’s systematic review, a large amount of empirical research has proved that transformational leadership can directly predict followers’ PWB [72]. Our study goes further, to reveal that in the educational context, transformational leadership can interact with teachers’ EI to jointly enhance their PWB.

Furthermore, the results of this study show that while PTL did not moderate the direct effect of teachers’ EI on OC, it did moderate the indirect effect linking teachers’ EI to OC via PWB. This is probably related to the mediating role of PWB, which not only manifests in the relationship between EI and OC, as shown in this study, but also in the relationship between transformational leadership and employees’ OC, as evidenced by prior studies. For example, Jain et al.’s study found that the positive effect of a leader’s transformational leadership on employees’ OC was mediated by employees’ PWB [73]. Moreover, ample empirical evidence has demonstrated that transformational leadership enhances employees’ OC through the mediating role of various other psychological and affective factors of employees, such as psychological empowerment [74], job satisfaction [75], and self-efficacy and autonomy [5,9]. Therefore, it can be explained that the interaction of PTL with teachers’ EI indirectly influences their OC through a mediation mechanism, and specifically through the mediating role of teachers’ PWB in this study. In essence, when teachers are emotionally intelligent and the principals in their schools perform transformational practices frequently, they enjoy improved emotional and psychological states, consequently bolstering their commitment to the school. Given the absence of prior research examining the moderating role of transformational leadership in the relationship among EI, PWB, and OC, our findings warrant the need for further investigations in this domain.

## 5. Implications

The findings of this study provide several practical implications for educational administrators and practitioners seeking to improve teachers’ OC and address the prevalent challenges of teacher attrition and shortages of qualified teachers.

Firstly, the results suggest that teachers’ EI plays a salient role in enhancing their OC, both directly and indirectly through the mediating pathway of PWB. This highlights the necessity for teachers to possess high levels of EI. In light of the “affective revolution” and the global education trend of Social Emotional Learning, which is largely grounded in the theory of EI, there is growing attention being placed on the importance of nurturing students’ emotional abilities [76]. However, targeted training for teachers’ EI remains insufficient [19]. To help teachers deal with increasing educational demands and stay committed and efficacious, we strongly recommend integrating tailored EI enhancement into teachers’ professional development programs. These programs should incorporate emotion-related knowledge, enabling teachers to enhance their awareness of both positive and negative emotions, and provide them with multiple chances to practice emotional intelligence competencies in their daily interactions.

Secondly, our study finds an intervention mechanism whereby teachers’ PWB mediates the influence of teachers’ EI on their OC, implying the importance of teachers’ PWB in determining their commitment levels. PWB, indicative of positive functioning and flourishing in life [77], is largely dependent on one’s perception of autonomy, environmental mastery, relations with others, etc. [30]. Therefore, educational administrators and principals should adopt multiple strategies to create warm, supportive, and encouraging work environments in the school, in order to improve teachers’ PWB [78]. Moreover, theme-specific training programs focusing on stress management can be introduced within schools to develop teachers’ abilities to cope with pressure and to motivate them and retain their sense of PWB. In this respect, EI-related training programs such as the “Cultivating Awareness and Resilience in Education (CARE)” program [79], which was developed specifically for teachers and has proven to be beneficial for trainees’ emotional ability and well-being, could be considered.

In addition, this study reveals that PTL plays a positive moderating role, interacting with teachers’ EI in predicting desired outcomes. Specifically, in schools led by transformational leaders, the positive effects of teachers’ EI on PWB and OC are amplified. Therefore, when considering new measures to increase teachers’ OC, enacting transformational leadership from principals represents a promising approach. To optimize the effectiveness of PTL, principals should undergo specified training programs [80], which will equip them with effective leadership behaviors. These includes awakening, attending to, and stimulating the high-level needs of teachers; assisting teachers to achieve self-realization by giving them autonomy and appropriate authorization; aligning the school’s vision with teachers’ personal goals, and experiencing the value of work in the process of self-realization. By adopting these leadership behaviors, principals can facilitate teachers’ self-realization and cultivate a sense of well-being and commitment to the school and the teaching profession.

## 6. Limitations

When interpreting our findings, several limitations should be considered. Firstly, the cross-sectional design limits the ability to draw causal inferences, especially in terms of the relationship between PWB and OC, which are believed to be interrelated to and strengthen each other. Although Yin et al.’s longitudinal study supported the notion that happy teachers are committed, but not vice versa [70], there exists some evidence pointing in the opposite direction, where OC predicts PWB [81,82]. Therefore, future research is encouraged to test our model using tracking data or experimental designs to ascertain causality. Secondly, all the data were self-reported by teachers at the same time, although the result revealed that the common method bias did not affect this study. Future studies could attempt to use data from multiple sources (e.g., principals and students) and collect data over multiple time points. A third limitation is that our model was largely restricted to the positive effect of teachers’ EI on OC, mediated by teachers’ PWB and moderated by PTL. There are likely to be other constructs that influence these relationships. For example, optimism [31] and perceived stress [69] have been identified as mediators between EI and PWB, which may constitute a sequential mediating path linking EI to OC. Moreover, other contextual variables such as different leadership types, organizational climate, and social support could serve as moderators in our model to further elucidate the boundary conditions affecting the impact of EI on individual outcomes. Last but not least, broader policies and other external factors, such as high-stakes testing and limited teacher professional autonomy, which significantly impact teachers’ OC and lead to high rates of teacher attrition, were not included in this study. These factors warrant more focused attention in future research.

## 7. Conclusions

This research, grounded in the broaden-and-build theory of positive emotions and the trait activation theory, advances our empirical understanding as to how and when teachers’ EI influences their OC. This is particularly valuable in light of the global, severe issue of ongoing teacher attrition and outflow. In conclusion, our findings suggest that teachers with higher EI tend to be more committed to their schools, partially because they can reach high levels of PWB due to their EI-related abilities. Moreover, when teachers work in schools with a more transformational principal, the positive effect of teachers’ EI on PWB and sequentially on OC is more significant compared to schools led by principals who perform transformational practices randomly. Therefore, our study underscores the importance of strengthening teachers’ EI, improving their PWB, and promoting principals’ transformational leadership concepts and behaviors within schools.

## Figures and Tables

**Figure 1 behavsci-14-00345-f001:**
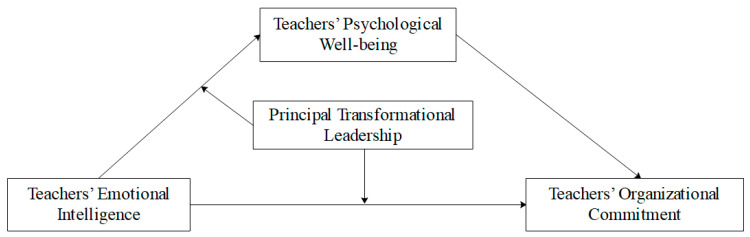
Hypothesized research model.

**Figure 2 behavsci-14-00345-f002:**
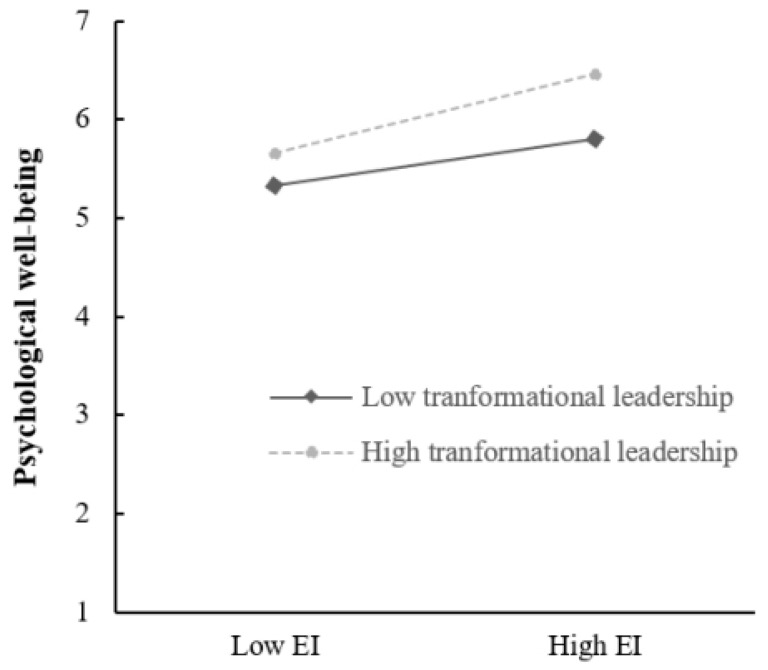
The moderating role of PTL in the relationship between EI and PWB.

**Table 1 behavsci-14-00345-t001:** Comparison of alternative models.

	χ^2^/d*f*	RMSEA	CFI	TLI	SRMR
Four-factor model	3.48	0.06	0.95	0.95	0.05
Three-factor model	8.04	0.10	0.87	0.85	0.11
Two-factor model	20.47	0.16	0.62	0.57	0.17
One-factor model	46.51	0.25	0.49	0.43	0.16

Note: three-factor model (PWB and OC combined); two-factor model (PWB, OC, and PTL combined); one-factor model (all variables combined). RMSEA = root-mean-square error of approximation; CFI = comparative fit index; TLI = Tucker–Lewis index; SRMR = standardized root-mean-square residual.

**Table 2 behavsci-14-00345-t002:** Descriptive statistics and Pearson correlation coefficients for the study variables (N = 768).

	1	2	3	4
1. EI	-			
2. PWB	0.50 **	-		
3. PTL	0.54 **	0.45 **	-	
4. OC	0.40 **	0.40 **	0.46 **	-
Mean	4.21	5.89	4.50	3.46
SD	0.53	0.87	0.66	0.58

Note: ** *p* < 0.01.

**Table 3 behavsci-14-00345-t003:** Mediation model test (N = 768).

Regression Equation		Fitting Index			Coefficient Significance		
Result Variables	Predictive Variables	R	R^2^	F	β	se	*t*
PWB	Gender	0.51	0.26	53.46 ***	0.20	0.07	2.71 *
	Teaching experience				0.00	0.00	0.60
	Educational background				0.08	0.07	1.09
	Professional rank				0.01	0.04	0.16
	EI				0.81	0.05	15.83 ***
OC	Gender	0.49	0.24	39.11 ***	−0.19	0.05	−3.91 ***
	Teaching experience				0.00	0.00	0.46
	Educational background				0.07	0.05	1.37
	Professional rank				0.06	0.03	2.24 *
	EI				0.28	0.04	7.11 ***
	PWB				0.19	0.02	7.75 ***

Note: non-standardized coefficients are reported. * *p* < 0.05, *** *p* < 0.001.

**Table 4 behavsci-14-00345-t004:** Decomposition of total effect, direct effect, and mediating effect.

	Effect Value	Boot se	Lower LCI	Upper LCI	Relative Effect Value
Total effect	0.44	0.04	0.37	0.51	
Direct effect	0.28	0.04	0.20	0.36	63.64%
The mediating effect of PWB	0.15	0.03	0.10	0.21	34.10%

**Table 5 behavsci-14-00345-t005:** Moderated mediation model test (N = 768).

Regression Equation		Fitting Index			Coefficient Significance		
Result Variables	Predictive Variables	R	R^2^	F	β	se	*t*
PWB	Gender	0.57	0.33	53.46 ***	0.21	0.07	3.066 ***
	Teaching experience				0.00	0.00	0.54
	Educational background				0.04	0.07	0.54
	Professional rank				0.00	0.04	−0.60
	EI				0.62	0.06	10.686 ***
	PTL				0.43	0.05	8.586 ***
	EI × PTL				0.27	0.05	5.226 ***
OC	Gender	0.54	0.29	38.42 ***	−0.19	0.05	−3.86 ***
	Teaching experience				0.00	0.00	0.43
	Educational background				0.04	0.05	0.76
	Professional rank				0.05	0.03	2.01 *
	EI				0.16	0.04	3.73 ***
	PWB				0.14	0.02	5.72 ***
	PLT				0.25	0.04	7.09 ***
	EI × PTL				0.01	0.04	0.41

Note: non-standardized coefficients are reported.* *p* < 0.05, *** *p* < 0.001.

**Table 6 behavsci-14-00345-t006:** Mediating effects at different levels of PTL.

	PTL	Effect Value	Boot se	Lower LCI	Upper LCI
The mediating role of PWB	M − 1 SD	0.06	0.02	0.03	0.11
	M	0.09	0.02	0.05	0.13
	M + 1 SD	0.11	0.02	0.06	0.15

## Data Availability

The datasets generated and/or analyzed during the current study are available from the corresponding author on reasonable request.

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
