# Peer review of "Teachers’ Emotional Intelligence and Organizational Commitment: A Moderated Mediation Model of Teachers’ Psychological Well-Being and Principal Transformational Leadership"

_behavsci, 2024, doi:10.3390/bs14040345_

Round 1

Reviewer 1 Report

Comments and Suggestions for Authors

This is a strong manuscript that makes a significant contribution to the literature. The authors offer a model for examining teacher attrition, critical given global teacher shortages. 

Author Response

Dear Reviewer,

Thank you for your encouraging and constructive comments regarding our manuscript. We are grateful for your recognition of the contribution our study makes to understanding teacher attrition, especially in the context of global teacher shortages.

Your positive feedback motivates us to continue our research and further refine our models and approaches. Thank you once again for your effort and thoughtful review.

Best,

Li, Mingwei; Liu, Feifei; Yang, Chuanli

Reviewer 2 Report

Comments and Suggestions for Authors

This article addresses an issue of great originality in educational research by linking teachers' emotional intelligence with what the authors refer to as "organizational commitment," which is particularly relevant to the practical reality of schools.

In this sense, the problem it addresses, the causes and consequences of the current instability affecting the teaching profession, are adequately described in the introduction. The key concept of organizational commitment is explained in this section, and the importance of teacher well-being is justified.

Initially, it may seem that the factor of "Principal Transformational Leadership" might require independent study, but the article adequately justifies the connection between this factor and the emotional dimension of teachers.

The methodology used is relevant and perfectly described.

The discussion of the results is well-articulated and allows for understanding the contribution that this study represents in relation to the conclusions of other authors. Similarly, the limitations indicated demonstrate the scope for further exploration of the topics addressed. It is recommended to include a reference to the need to compare quantitative results with those of a qualitative nature, obtained, for example, through narrative methodologies. An example would be this work: Luna, D., Pineda-Alfonso, J. A., García-Pérez, F. F., & Leal da Costa, C. (2022). Teacher uneasiness and workplace learning in Social Sciences: towards a critical inquiry from teachers’ voices. Education Sciences, 12(7), 486. https://doi.org/10.3390/educsci12070486

In conclusion, this is a well-conceptualized and developed study that can be published in its current version.

Author Response

Thank you very much for taking the time to review this manuscript. Please find the detailed responses below and the corresponding revisions highlighted in red in the re-submitted files.

Point-by-point response to Comments and Suggestions

Comments 1: It is recommended to include a reference to the need to compare quantitative results with those of a qualitative nature, obtained, for example, through narrative methodologies. An example would be this work: Luna, D., Pineda-Alfonso, J. A., García-Pérez, F. F., & Leal da Costa, C. (2022). Teacher uneasiness and workplace learning in Social Sciences: towards a critical inquiry from teachers’ voices. Education Sciences, 12(7), 486. https://doi.org/10.3390/educsci12070486

Response 1: Thank you very much for your suggestions. The qualitative work you recommended has been enlightening, helping us to understand the underlying causes of issues such as teacher turnover from the teachers' perspective.

Therefore, we have included it in the discussion section (page 9, line 322). Thank you very much. Below is the exact updated text in the manuscript for your reference.

“Moreover, challenges such as the phenomenon of “accountability”, daily overexertion, no time and classroom management exacerbate emotional demands on teachers, creating substantial pressure and potentially leading to resignations after wear and tear [68].”

Reviewer 3 Report

Comments and Suggestions for Authors

The manuscript is well-written and addresses a topic of on-going significance - teacher attrition.  Drawing our attention to the importance of EI and the intervening moderator of transformational leadership, findings help teacher educators and policymakers recognize the importance of preparing future educators for the emotional demands of our profession. One recommendation for the final submission is an addition to the limitations section. There are broader policy factors and other working conditions that greatly impact a teacher's decision to leave the profession; yet the author(s) do not acknowledge these factors as playing an equally powerful role in high teacher attrition.  EI and transformational leadership cannot fully mitigate the negative impact of these dynamics. It is important to acknowledge that we need to change working conditions (decrease high stakes testing,; restore professional autonomy, etc.) as part of our efforts to decrease teacher attrition and not solely focus on individual factors like an educator's EI.  Adding a few sentences to the introduction and the limitations acknowledging this reality would helpful.

Author Response

Thank you very much for taking the time to review this manuscript. Please find the detailed responses below and the corresponding revisions highlighted in red in the re-submitted files.

Point-by-point response to Comments and Suggestions

Comments 1: One recommendation for the final submission is an addition to the limitations section. There are broader policy factors and other working conditions that greatly impact a teacher's decision to leave the profession; yet the author(s) do not acknowledge these factors as playing an equally powerful role in high teacher attrition.  EI and transformational leadership cannot fully mitigate the negative impact of these dynamics. It is important to acknowledge that we need to change working conditions (decrease high stakes testing,; restore professional autonomy, etc.) as part of our efforts to decrease teacher attrition and not solely focus on individual factors like an educator's EI. Adding a few sentences to the introduction and the limitations acknowledging this reality would helpful.

Response 1: Thank you very much for your suggestions. The broader policy factors and other working conditions you mentioned, such as high-stakes testing and limited professional autonomy, are indeed important external factors that negatively affect teacher organizational commitment and turnover rates.

Therefore, we have followed your advice and added a few sentences in the introduction (page 1, line 39) and the limitations section (page 11, line 445). We hope this will help acknowledge this reality

Below is the exact updated text in the manuscript for your reference.

Introduction (page 1, line 39):

 “Contemporary teachers confront a complex and dynamic reality characterized by various demanding situations, such as high-stakes testing, excessive workloads, limited professional autonomy, and scrutiny from parents and society.”

Limitation (page 11, line 445):

“Last but not least, broader policy and other external factors, such as high-stakes testing and limited teacher professional autonomy, which significantly impact teachers’ OC and lead to high rates of teacher attrition, were not included in this study. These factors warrant more focused attention in future research.”